# A Particular Case of Urban Sustainability: Comparison Study of the Efficiency of Multiple Thermal Insulations for Buildings

Simon Pescari [1], Mircea Merea [1], Alexandru Pitroacă [1] and Clara-Beatrice Vilceanu [2,*]

1. Department of Civil Engineering and Building Services, Politehnica University Timisoara, 300223 Timisoara, Romania
2. Department of Overland Communication Ways, Foundations and Cadastral Survey, Politehnica University Timisoara, 300224 Timisoara, Romania
* Correspondence: beatrice.vilceanu@upt.ro

**Abstract:** Achieving urban sustainability is a complex process that includes improving buildings' resilience and energy efficiency by using the optimum thermal insulation solution. With the advent of new energy restrictions, it is very important to find the best compromise between the price of the thermal insulation material and energy savings because, sometimes, the initial cost of a thermal rehabilitation seems to be very high. The purpose of this study is to illustrate the variations in the amount of heat energy required by a multi-storey residential structure in Romania that uses 14 various kinds of thermal insulation materials. The energy demand is determined using the dynamic method using a building energy simulator that can evaluate the energy usage of lighting, warmth, ventilation, climate control, and water heating.

**Keywords:** thermal insulation; dynamic method; thermal resistance; energy consumption; urban sustainability

## 1. Introduction

The building sector has a major impact on the environment; almost 30% of the $CO_2$ emissions are generated by this sector globally. The energy consumption of the building sector in the EU is around 40%. One solution to reducing these values in a sustainable way is by increasing the building's envelope efficiency. The development and use of insulation materials are two of the most significant energy-saving strategies [1]. Using the optimum thermal insulation solution for improving buildings' resilience and energy efficiency [2] contributes to achieving urban sustainability in the actual context of climate change.

There are various solutions for thermal insulation. The insulation material resources are dominated by fossil resources, followed by mineral resources, and just a small part by renewable resources (about 9% in 2019 in Germany) [3]. Each material has its advantages and disadvantages, and a proper solution must be found. This study makes a comparison between 14 different insulation materials, considering the material's thermal efficiency and the life cycle cost of its usage.

When considering the usage of a material, an important decision factor must be the environmental impact. The $CO_2$ emissions must be taken into account when choosing the insulation material [4]. When considering a low-impact solution due to the sequestration of carbon, a good solution is wood-fibre board, a study shows. The material could be recycled if it is uncontaminated. For example, instead of using EPS, using wood fibre for 5 million houses would avoid 3 million tonnes of $CO_2$ emissions [5]. In addition, in the same study, it is shown that among the following materials: EPS, mineral wool, and phenolic foam, EPS has the lowest environmental impact. Another study that examined the life cycles of eight different types of insulation came to the conclusion that foam glass had the highest life cycle emission of carbon dioxide, while phenol formaldehyde had the lowest [6].

Sustainable solutions should be taken into account. If it can be manufactured at a scale that is economically viable and sustainable, sheep wool insulation can compete with the forms of insulation that are currently on the market [7]. It represents a sustainable and innovative method regarding waste of local low-quality wool [8].

The thickness of the insulation is also a debatable point. The insulation thickness has a very big impact on thermal comfort. Since it greatly increases the risk of overheating, super insulation may be harmful in the summer. A solution to this problem may be a ventilated external insulation layer [9]. Because of the tightening regulations regarding the minimum thermal resistance, the insulations used must be thicker and thicker (e.g., 20–25 cm in Hungary and about 40–50 cm in the Nordic countries). The solution would be more efficient insulation, with a conductivity level of less than 0.020 W/mK. Some materials with these properties are vacuum insulation panels (VIP) or aerogels. A 3 cm layer with VIP would be equivalent to a 10 cm layer with EPS [10]. On the negative side, these techniques will never completely solve the problems of brittle structures (aerogels) and vacuum loss (VIP). Making a material containing nanopores using a controlled technique may represent a solution [11].

Increasing the thermal insulation thickness may not always be a solution. For example, increasing the straw insulation thickness from 10 cm to 20 cm does not have any significant effect on the interior air temperature [12].

A solution for reducing the insulation cost may be to use a different thickness on every wall orientation. A study shows that for the best life cycle cost, the insulation thickness may vary depending on the wall orientation, the insulation on the north side may be thinner than the one on the west or east side (the study is realised for heating and cooling in Morocco) [13].

In this study for every material, the comparison is made with a 10 cm insulation layer applied on the exterior surface. The same thickness was used to highlight the differences between the thermal resistance of the materials.

Sound insulation is not covered in this article but is also a factor that should be taken into account when choosing the insulation material. A study shows that using mineral wool instead of EPS increases the sound insulation. EPS has a negligible influence on sound insulation. In addition to sound insulation, a plaster layer could be used, increasing the sound insulation from 3 dB to 7 dB but having no effect on the thermal insulation [14].

The thermal transmittances (U-values) used in this study are the values offered by the material producer. A study comparing hemp and stone wool insulation showed that the results of the in-situ measurements regarding the real U-value were slightly lower than the U-values derived from numerical calculations [15]. The difference would not significantly affect the heat necessary demand values.

## 2. Materials and Methods

This study is conducting a comparison between different insulation materials, focusing on the energy efficiency of each material and the life cycle cost of its usage. The efficiency of the materials is determined by using the dynamic method implemented within the EnergyPlus software.

This study is realised by comparing the heat necessary demand on a building located in the city of Timisoara ($45.780118428000584°$ N, $21.234839680908596°$ E), in the western part of Romania, using each material studied as an insulation solution. The solutions are applied only to the exterior walls. The building was constructed before 1989, and it is made of concrete panels with almost no insulation solution applied at that time. The building has five stories and no framing structure. This type of building was common in the communist era, and the solution deducted in this article may be used on each and every one of them. The building is located in climate zone II for heating and climate zone III for cooling. In Romanian normative C107/3-2005, five climate zones for heating calculus and four climate zones for cooling calculus are defined. The building's current floor plan is shown in Figure 1.

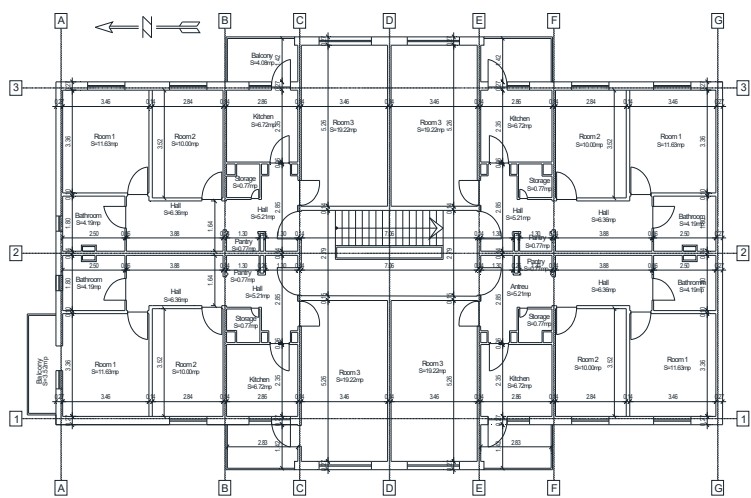

**Figure 1.** Building floor plan.

The exterior wall is composed of four layers: a structural layer on the inside of 125 [mm] (reinforced concrete), an insulation layer of 85 [mm] (AAC), a protection layer of 60 [mm] (reinforced concrete), and a cement mortar layer of 10 [mm] (Figure 2).

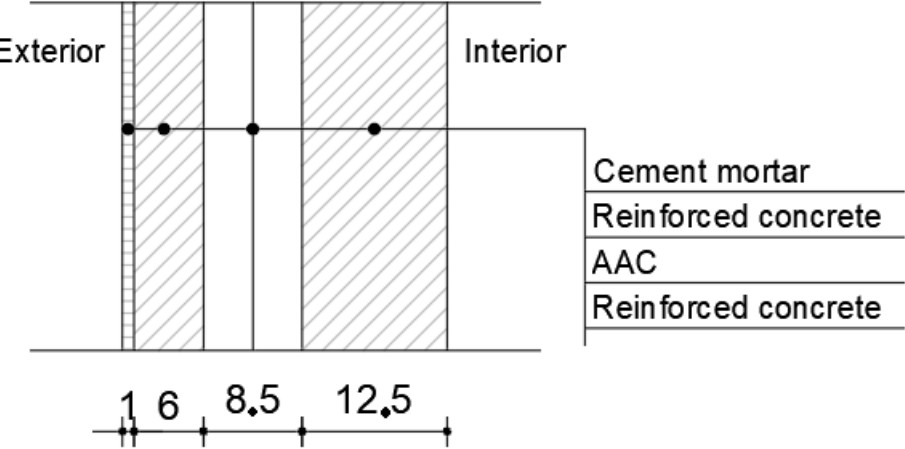

**Figure 2.** Exterior wall layer.

The life-cycle cost may be the most important factor when choosing the insulation material. Considering the fact that a thermal insulation solution must be applied to the majority of the buildings, the cost of this process is a key factor because the budget is limited. The life cycle cost analysis is made using an Excel file developed by this study team.

## 3. Thermal Insulation Solutions

This study compares 14 different insulation materials. The comparison is focusing on the thermal conductivity and the life-cycle cost. Even though not every material can be used on an exterior wall, this study focuses on the material's efficiency. From the studied materials, flexible wood fibres are used for roof insulation; cellulose is used as interior insulation, for roofs or attics, because it is not suitable for high-humidity places; straw can be used for exterior walls, but only if it has been applied for humidity protection; glass mineral wool is used for plane surfaces; and extruded polystyrene is used for foundations because it is an impervious material and does not let the interior humidity escape. Each material compared in this study is briefly described as follows. Each material's characteristics are described in Table 1.

**Table 1.** Thermal insulation material's characteristics.

| Material | Thermal Conductivity [W/Mk] | Thermal Resistance (For 100 mm) [Km$^2$/W] | Density [kg/m$^3$] | Solution Number |
|---|---|---|---|---|
| Rigid wood fibers | 0.038 | 2.5 | 160 | S1 |
| Flexible wood fibers | 0.038 | 2.6 | 50 | S2 |
| Cellulose | 0.035 | 2.632 | 27–65 | S3 |
| Wool | 0.038 | 2.63 | 23 | S4 |
| Hemp | 0.039–0.040 | 2.5 | 25–38 | S5 |
| Hempcrete | 0.06 | 1.429 | 275 | S6 |
| Cellular glass | 0.041 | 2.439 | 115 | S7 |
| Straw | 0.08 | 1.248 | 110–130 | S8 |
| Glass mineral wool | 0.035 | 2.85 | 20 | S9 |
| Mineral stone wool | 0.032–0.044 | 2.7–2.85 | n/a | S10 |
| Polyurethane foam | 0.023–0.026 | 4.5 | 30–40 | S11 |
| Expanded polystyrene | 0.034–0.038 | 3.52 | 15–30 | S12 |
| Extruded polystyrene | 0.033–0.035 | 3 | 20–40 | S13 |
| Aerogel | 0.014 | 7.6 | 150 | S14 |

### 3.1. Wood Fibre

Wood-fiber insulation was initially launched about 20 decades ago after experts working in European timber manufacturing locations discovered innovative ways to turn the waste wood. There are two types of wood fibres: rigid wood fibres and flexible wood fibres.

### 3.2. Cellulose

The newspaper that has been recycled is used to produce cellulose. The paper is grated, and inorganic salts such as boric acid are added to make it resistant to fire, mould, insects, and parasites. Depending on the application, the insulation is applied either by blowing it or by spraying it with moisture.

### 3.3. Wool

Wool insulation is manufactured from sheep's wool fibers that are either mechanically bound together or attached with recycled polyester adhesive at a rate of between 5% and 15% to create insulating strips and rolls. The sheep are not grown in the first place for their wool; however, they must be cut down for their health. The wool used for realising insulation is discarded wool from other industries because of its colour or quality.

### 3.4. Hemp

Hemp fibres are realised from the hemp plant's straws. The hemp grows to a 4 m height in a 100–120 days period. Because of the fact that plants are shadowing the soil, chemical protection or harmful additives are not required. The product is usually made with 85% hemp fibres and 3–5% chiffon added for fire protection.

### 3.5. Hempcrete

Hemp and lime (possibly even hydraulic lime, sand, and cement) are combined to create hempcrete, which is used as insulation and building material. Hempcrete is a material that is simple to use. There are no expansion joints necessary since the substance is not fragile like concrete. Hemp walls must be used in conjunction with a framework made from alternative material that can sustain the vertical load because the density of the hemp cement is only 15% of the traditional concrete.

### 3.6. Cellular Glass

Cellular glass is manufactured for the most part from recycled glass and mineral base materials, such as sand. The mixture is melted into glass, which is then cooled and turned into a fine powder. Glass in the form of powder is cast into the heated matrix in

a "synthesizing" process. Following the addition of a small quantity of finely powdered carbon black, the material is heated during the "celling" process. The insulating bubbles are produced when the carbon and oxygen react, forming carbon dioxide. Carbon dioxide represents more than 99% of the gas from the cellular spaces.

### 3.7. Straw

Straws are a secondary agricultural product made from the cereal plant's dried steams after the grains and chaffs are removed. The straw represents approximately half of the cereal culture yield, such as barley, oats, rice, rye, and wheat.

### 3.8. Glass Mineral Wool

Manufactured with melted glass, usually with 20% to 30% recycled industrial waste. The material is made with glass fibres organised using a binder with a similar texture to wool. The process captures a lot of air gaps in the glass, and these air gaps have high properties for thermal insulation. The material density may vary by pressure and binder content.

### 3.9. Mineral Stone Wool

Mineral stone wool is a product made with melted rock at a temperature of approximately 1600 °C, where either air or steam is blown. With a typical diameter of 2 to 6 μm, the final product is a mass of tiny, knitted fibers. A polymer binder and oil may be included in mineral wool to serve as a dust reducer.

### 3.10. Polyusicyanide Foam/Polyurethane Foam (PIR/PUR)

The polymer known as polyurethane (PUR and PU) is made up of two organic units connected by carbamate bonds (urethane). A range of densities and hardnesses, including isocyanate, polyol, or additives, can be used to produce polyurethane. PIR, commonly known as polyisocyanurate, is a thermoresistant polymeric substance often used as foam for rigid thermal insulation. Its chemistry is comparable to that of polyurethane (PUR), with the exception of the larger amount of methylene diphenyl diisocyanate (MDI) and the use of a polyester-derived polyol in the reaction rather than a polyether polyol. Additionally, the catalysts and additives used for PIR are different from those used for PUR. Precast PIR sandwich panels with corrugated steel facades are created, protected from corrosion, bonded to a PIR foam core, and extensively utilized as insulation for roofs and vertical walls (for example, for deposits, factories, office buildings, etc.).

### 3.11. Expanded Polystyrene

A synthetic aromatic polymer made from styrene monomer is called polystyrene. Polystyrene comes in foam or solid forms. Closed-cell foam with a hard and rough surface is known as expanded polystyrene (EPS). Usually, it is white and is manufactured with pre-expanded polystyrene beads. Polystyrene is one of the most used plastic materials, and the scale of production is a few billion kilograms per year.

Polystyrene foams are manufactured using sulphating agents, which form bubbles and expand the foam. These agents are usually hydrocarbons, such as pentane.

Both expanded polystyrene and extruded polystyrene are closed-cell foams, although they are not totally impermeable or water-resistant. Thrown polystyrene does not biodegrade for hundreds of years, and it is resistant to photolysis.

### 3.12. Extruded Polystyrene

Extruded polystyrene foam (XPS) has closed cells, increasing the surface roughness while also increasing stiffness and decreasing heat conductivity. It is a little denser and so a little harder than EPS.

The diffusion resistance to steam of XPS is very low, and that makes it proper for usage in humid environments.

### 3.13. Aerogel

Aerogel is a synthetic material that is extremely lightweight and porous and is made from a gel in which the liquid portion has been swapped out for gas. A solid with extremely low density and low thermal conductivity is the final product. The material feels similar to fragile expanded polystyrene. Aerogels may be manufactured from a variety of chemical cops.

Aerogels are effective thermal insulators because they eliminate two of the three methods of heat transmission (convection, conduction, and radiation). Because they are nearly completely made of gas, which conducts heat poorly, they are excellent conductive insulators. Because the air cannot enter through the grid, aerogels are convective inhibitors. Because they are penetrated by infrared radiations, which transfer heat, aerogels are poor radiative insulators. The most popular type of aerogel is the one that contains silica. Silica does solidify in dimensional clusters between them, which contain only 3% of the volume. So, the conductivity through the solid is very low. The rest of 97% is composed of air in small nanopores. The air has almost no movement space, inhibiting convection and conductivity in the gas phase.

## 4. Heat Necessary Demand

The energy demand is determined using the dynamic method. The dynamic method calculation is realized using the EnergyPlus specialized software. The software represents a building energy simulator that can evaluate the energy usage of lighting, warmth, ventilation, climate control, and water heating. Data input referring to the geometry of the building is realized by manually introducing each element in a global coordinate system. In this study, for the building modelling, another specialized software is used, namely SketchUp. Additionally, thermal zones are made using the OpenStudio program to divide interior spaces from outdoor spaces. For exporting the model to EnergyPlus, an extension from the OpenStudio software is used. After importing the file and introducing the climate data from a file that contains the climate data of the studied zone, a simulation is realized. The two additional programs (SketchUp and OpenStudio) were used because the EnergyPlus drawing module uses only spatial coordinates, so SketchUp was used for a more user-friendly interface, and for the same principle OpenStudio was used for modelling the thermal zones.

Using the OpenStudio software, there are realized thermal zones for the interior spaces. For each floor, a thermal zone is assigned, with the interior slab defined as an interior element. The balcony is defined as a shading element of the neighbouring buildings (Figure 3). Because the two buildings are connected and it is assumed that the temperature between them is the same, the envelope element between the analysed building and the southern building is defined as adiabatic (Figure 4). Between the analysed building and the northern building, there is an air gap, so the envelope element between them is considered to be exterior but is not affected by solar radiation or wind.

From OpenStudio software, the file is exported into an accepted format by EnergyPlus.

Because the EnergyPlus software does not take into consideration the increased coefficient of thermal conductivity, there have been used increased values of the thermal conductivity. According to the Romanian methodology Mc001/2017, the thermal conductivity should be amplified with a correction factor (a) that takes into account the aging of the element.

Another characteristic that must be introduced in EnergyPlus is represented by internal gains. For the internal gains, there have been used internal gains from people and from lightning. It was realized that there was an occupancy program for the building and a usage program for the interior lighting. Every hour of the day, the program accounts for the building's occupancy rate as well as the amount of time that the lightning system is used.

The heating system, air cooling, and ventilation are considered to be an ideal system, where the losses from the installation equipment are not taken into account and the interior

temperature is constant for heating and cooling. The temperature considered for heating is 20 °C, and cooling is 25 °C, values recommended by ISO 13790.

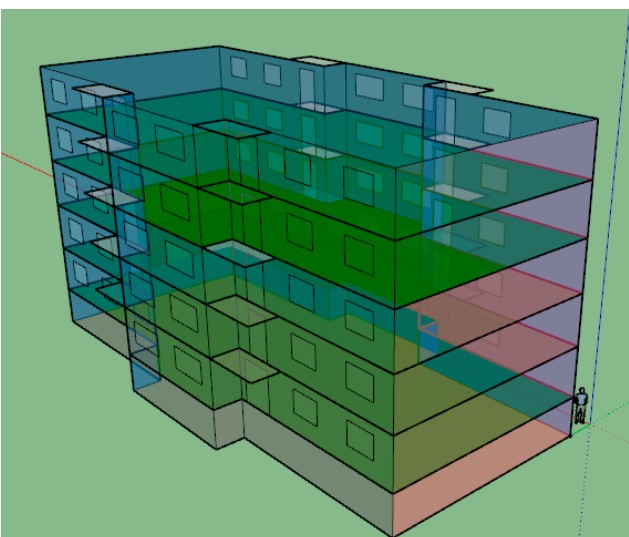

**Figure 3.** Thermal zones (green—delimitation of interior spaces; blue—delimitation of interior spaces from exterior spaces).

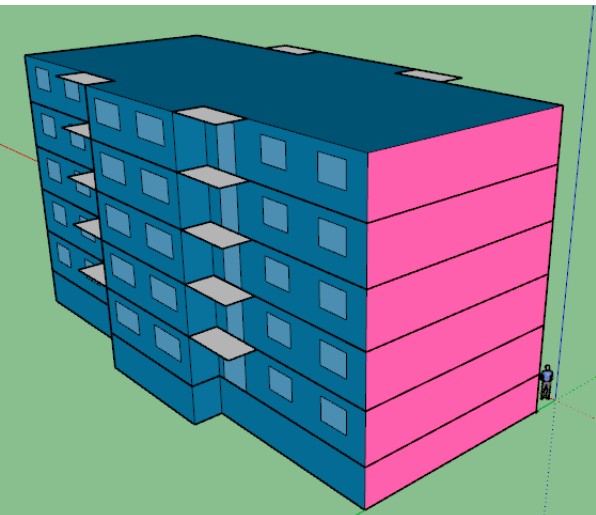

**Figure 4.** Thermal zones (pink—delimitation of adiabatic spaces; blue—delimitation of interior spaces from exterior spaces).

In addition, from the official webpage of EnergyPlus, the climate data for the location used (in the current study, Timisoara) is downloaded. The temperature values are obtained from temperature recordings every hour for an 18-year period (usually 1982–1999) shared by ASHRE (American Society of Heating, Refrigerating, and Air-conditioning Engineers). Characteristic data such as air temperature, relative humidity, atmosphere pressure, global solar horizontal radiations, normal direct solar radiations, wind speed, wind direction, and other derived values from the ones mentioned are contained in the weather files. From the EnergyPlus software, the building's orientation is defined so it respects the real-life building's orientation, and it is the same in each case.

The calculation was set to carry on for a year at a 6-min interval, with a ratio of 10 units (the values being between 1 and 60).

The first analysis was realised when considering the exterior walls without insulation. Then, for the other analyses, an extra layer was added with the material properties (rough-

ness, thickness, conductivity, density, and specific heat). In this study, for every material, the thickness is considered to be 100 mm, even if for some materials such as aerogel, such a thickness is not available. In the table below, the results are displayed. The table contains the material price (available for 2021), the material conductivity, the heat demand obtained using the dynamic method, and the heat reduction percentage compared to the case where no insulation material is used (Table 2).

**Table 2.** Thermal insulation materials solutions.

| Insulation Material | Price/sqm [€] | Conductivity [W/mk] | Heat Demand [kWh/m$^2$/Year] | Heat Reduction % |
|---|---|---|---|---|
| No insulation | - | - | 114.34 | - |
| Rigid wood fibres | 27.81 | 0.038 | 75.23 | 34.21 |
| Flexible wood fibres | 14.72 | 0.038 | 75.29 | 34.15 |
| Cellulose | 6.04 | 0.035 | 74.65 | 34.71 |
| Wool | 5.24 | 0.038 | 75.31 | 34.14 |
| Hemp | 16.47 | 0.04 | 75.73 | 33.77 |
| Hempcrete | 39.49 | 0.06 | 79.42 | 30.54 |
| Cellular glass | 28.28 | 0.041 | 75.92 | 33.60 |
| Straw | 3.20 | 0.08 | 82.50 | 27.85 |
| Glass mineral wool | 2.37 | 0.035 | 74.67 | 34.69 |
| Mineral stone wool | 6.74 | 0.04 | 75.75 | 33.75 |
| Polyurethane foam | 15.75 | 0.024 | 72.11 | 36.93 |
| Expanded polystyrene | 9.62 | 0.038 | 75.31 | 34.14 |
| Extruded polystyrene | 13.05 | 0.035 | 74.67 | 34.69 |
| Aerogel | 689.54 | 0.014 | 69.45 | 39.26 |

The heat demand reduction varies depending on the solution, from 27.85% (if using straw) to 39.26% (if using aerogel) (Figure 5).

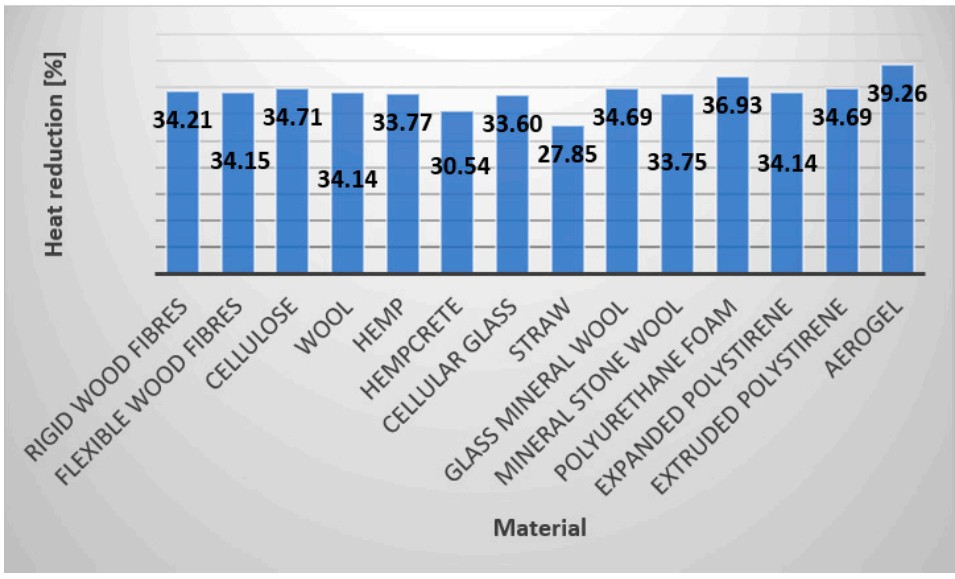

**Figure 5.** Heat reduction for each solution used.

The prices include only the material cost, without any binders or workmanship, as shown in Figure 6.

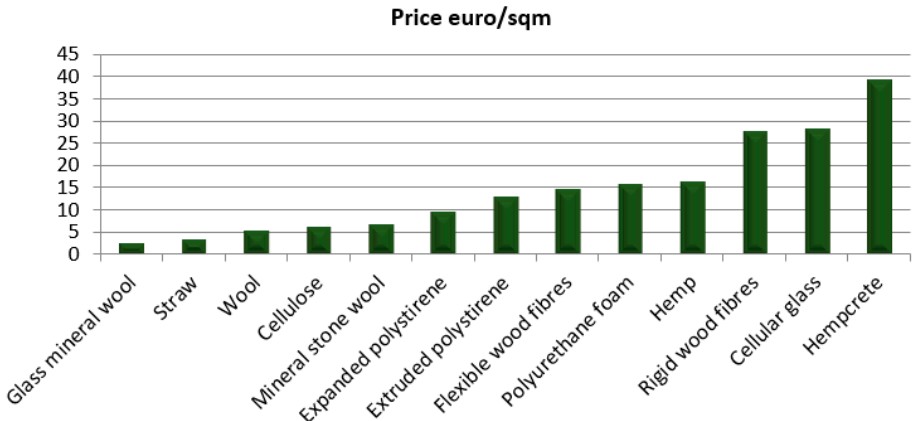

**Figure 6.** Price comparison of the materials (in euro/sqm).

The difference between the heat necessary demand of the building using different types of materials may not be very significant, but considering the energy price increase, any reduction can make a difference, as represented in Figure 7.

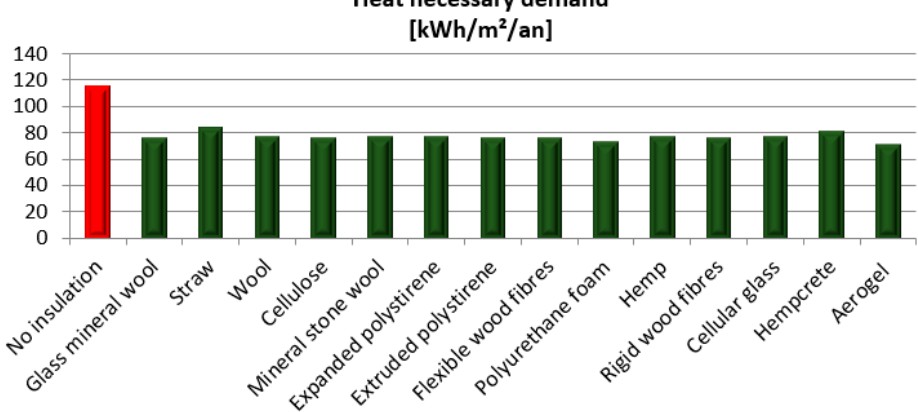

**Figure 7.** Heat necessary demand comparison of the enveloped building with each material.

## 5. Life Cycle Cost

Because of the high energy consumption of the building and the increasing cost of energy, a measure must be taken. The solution has to reduce the energy demand but also be cost-efficient. The life-cycle cost must be taken into account rather than focusing on the investment cost.

Life cycle cost evaluation is an economic method to evaluate a solution that takes into account not only the initial cost but also the further costs (maintenance, exploitation, and replacement).

The method is useful and is recommended for every capital investment that has a higher initial cost with the ulterior purpose of reduced cost.

The global cost for the analysed period is composed of the initial investment, the annual cost, the actualization factor, and the waste value for the analysed final period (Figure 8) (1).

$$C_g(\tau) = C_I + \sum_j [\sum_{t=1}^{\tau} (C_{a,i}(j) \times R_d(i)) - V_{f,\tau}(j)] \tag{1}$$

$C_g(\tau)$—global cost for analysed period;
$C_I$—initial investment;
$C_{a,i}(j)$—annual cost for year "i";
$R_d(i)$—actualisation factor;
$V_{f,\tau}(j)$—waste value for the analyse final period.

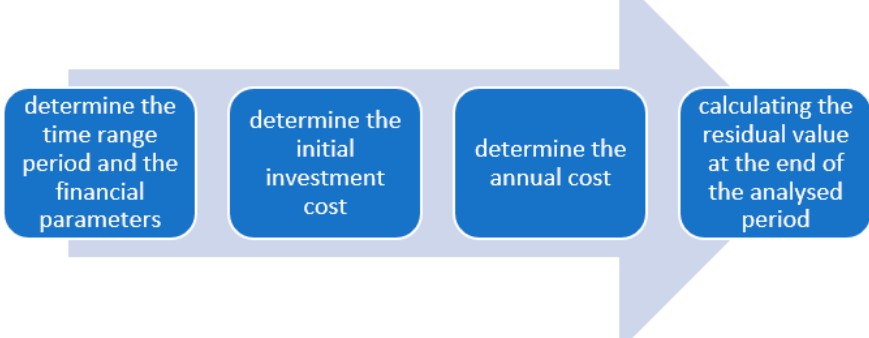

**Figure 8.** Steps for calculating the life cycle cost.

The first step in this analysis is to determine the time period and the financial parameters. The period may be shorter than the building's life cycle and takes into account the technical functioning period of the elements. In the global cost, the real annual cost growth rate is also considered, which excludes inflation.

The second step is to determine the initial investment cost. The cost includes all the costs that are summed up until the moment when the investment is handed over to the customer, ready to be used. This includes design, construction material acquisition, and workmanship.

The third step represents determining the annual cost. This cost includes the annual energy cost. It is determined considering the calculated building's energy consumption in the design phase, using the charge for each type of energy and considering the price fluctuation. Additionally, the annual cost includes the maintenance cost. These costs are necessary to maintain the building's systems within optimal parameters. These include annual costs for inspection, cleaning, adjustments, and repairs. This is determined based on Annex A of standard EN 15459:2007. Another cost included in this step is the periodic replacement cost. These costs are for replacing the elements that have a lower life period than the analysed period. If a component has a longer life period than the analysed period, the residual value of the component is calculated at the end of the period and it is decreased from the global cost.

The fourth step involves calculating the residual value at the end of the analysed period. It is composed of the sum of the values from each element. If the global cost period is different from the life period of a component, the last replacement cost must be taken into account when calculating the global cost. For example, for a component with a life period of 40 years and constant depreciation, the residual value after 30 years (the end of the analysed period) is 25% of the initial investment. This value must be actualised at the beginning of the analysed period.

For each thermal insulation solution described, a life cycle cost determination is realized. The return period on investment is between 2 years and 7.5 years, depending on the chosen solution. The last solution (with aerogel) is not shown in Figure 9 because the return period on investment is longer than 20 years (the life period of an insulation material). The chart also shows the importance of the investment cost, which is the key factor in choosing the solution rather than the material's thermal efficiency.

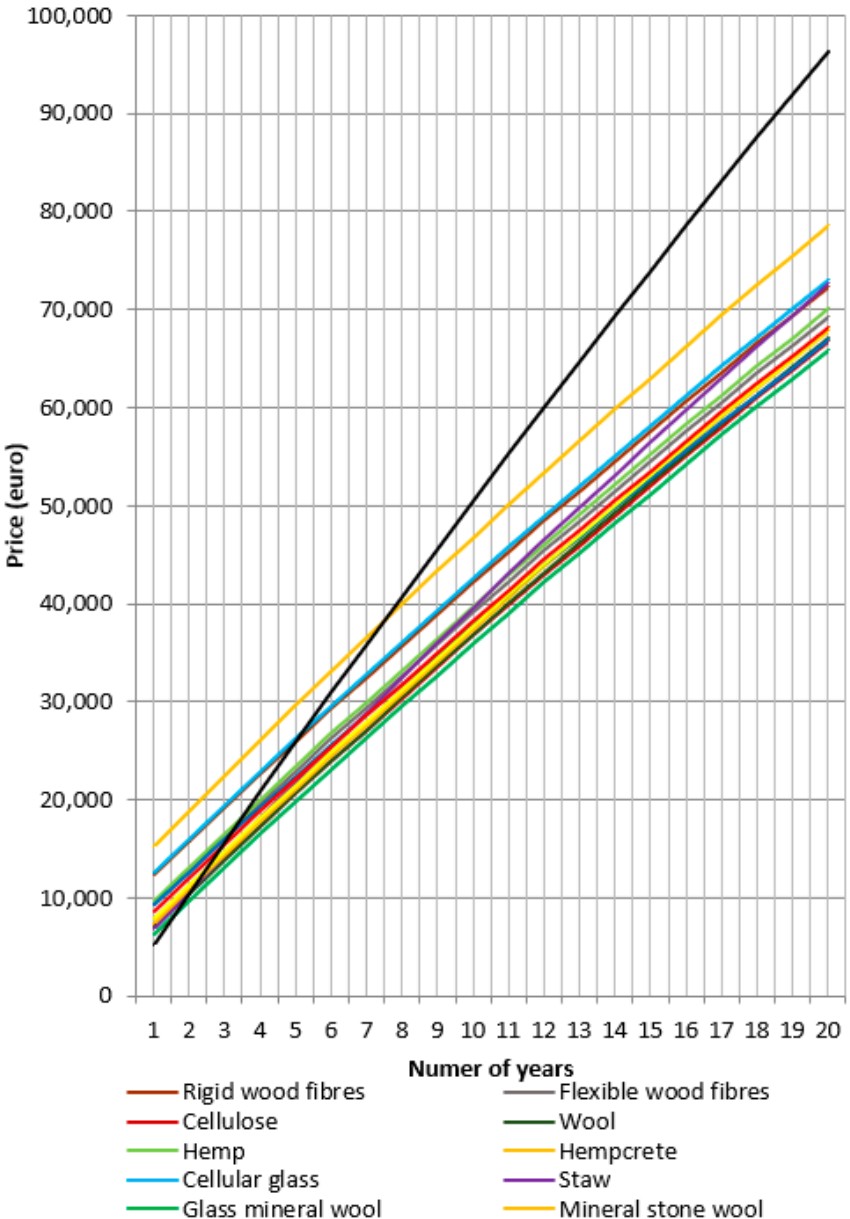

**Figure 9.** Life cycle cost for each solution used in this study.

This study shows that thermal conductivity is not such an important factor when choosing an insulation material. The life-cycle cost is the crucial factor when choosing a solution. For example, aerogel, the best thermal insulation material, is not an economically viable option at the moment because of its high cost (Figure 10).

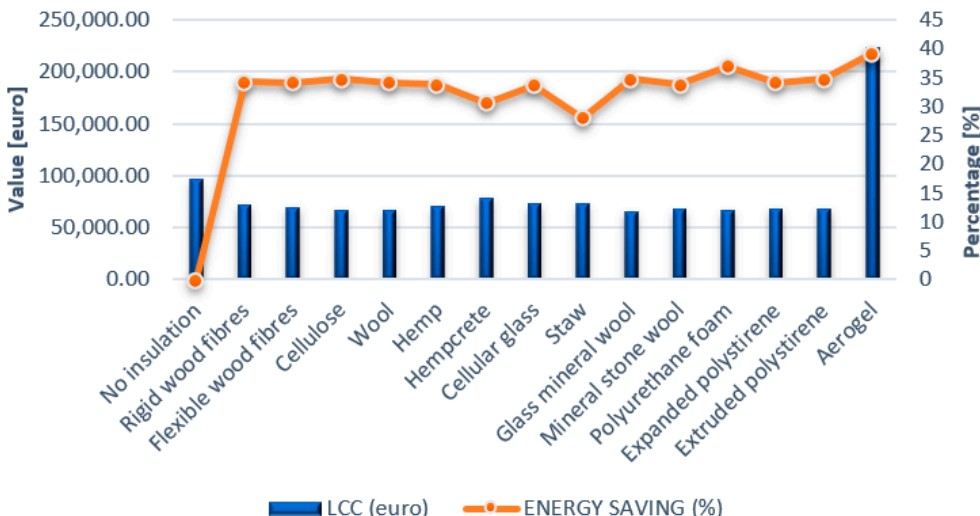

**Figure 10.** Comparison chart between the 14 solutions considering the heat reduction and the life cycle cost.

Considering the increasing price of energy, the return period on investment gets lower over the years. In this study, the most economical solution is when using wool (S4) as a thermal insulation material. The return period for this solution is about 2 years. It is a cheap and sustainable solution; it can be manufactured quite easily by using unwanted sheep wool. A downside of this material is that it cannot be obtained for high demand, so other solutions would be necessary. A more common solution with a low return period on investment is mineral stone wool, with a return period of a little more than two years. The material can support high demand and is also a sustainable material.

## 6. Conclusions and Discussions

Considering the European directives regarding the reduction of energy consumption, new and better solutions must be found and implemented. The first solution to take into consideration for reducing the energy consumption of a building is improving the envelope. In this study, the element studied is the exterior wall, which represents 50% of the building's envelope. Fourteen solutions were taken into account, but only ten can be used on the exterior walls (S1, S2, S4, S5, S6, S7, S8, S10, S12, and S14); the rest are only for comparison purposes. Of these ten solutions, only nine are profitable, reducing the total cost (S1, S2, S4, S5, S6, S7, S8, S10, and S12).

There is no perfect solution when considering such an investment, but almost every solution is profitable for the building's residents. This study shows only the criteria that should be considered when choosing the solution, and for every building there can be a different solution.

**Author Contributions:** All authors have contributed to the work as it follows: conceptualization, software, validation, writing—original draft preparation, M.M. and A.P.; writing—review and editing, project administration, C.-B.V.; and methodology and supervision, S.P. All authors have read and agreed to the published version of the manuscript.

**Funding:** This research received no external funding.

**Institutional Review Board Statement:** Not applicable.

**Data Availability Statement:** Not applicable.

**Acknowledgments:** This research did not receive any specific grant from funding agencies in the public, commercial, or not-for-profit sectors.

**Conflicts of Interest:** The authors declare no conflict of interest.

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
