# Peer review of "A Particular Case of Urban Sustainability: Comparison Study of the Efficiency of Multiple Thermal Insulations for Buildings"

_sustainability, doi:10.3390/su142316283_

Round 1

Reviewer 1 Report

The manuscript is original and has a good content with interesting analyses and discussion, therefore, being acceptable for publication. Minor English errors have been appointed in the annexed file, that can be improved. The contribution must also be enhanced in several other ways, i.e. in terms of figures, some extra tables and further clarification in many of the passages, all of them marked and explained in the annexed file as well. Such demands are all compulsory, and will considerably improve the understanding, hence visibility, of the paper.

Author Response

Thank you for your time to do the review and also thank you for your recommendations as they consist in objective ideas meant to improve our work. We took into consideration your suggestions and made appropriate changes to the article as you will see in the word document as we have used Track Changes function.

Every country has its on regulations regarding the minimum thermal resistance in the national codes. The values are cited from other articles.

To compare the thermal resistance between two materials, the same thickness must be used, because the most important insulation characteristic is the thermal conductivity. The thermal resistance is equal to the thermal conductivity divided by the thickness.

In the Romanian normative C107/3-2005 there are defined five climate zones for heating calculus and four climate zones for cooling calculus, each with its own external temperature for calculus.

The life cycle cost is detailed in chapter 5. The excel file follows the steps described there.

The two additional programs (SketchUp and OpenStudio) were used because, EnergyPlus drawing module uses only spatial coordinates, so SketchUp is used for a more user-friendly interface and for the same principle OpenStudio is used for modelling the thermal zones.

According to the Romanian methodology Mc001/2017 the thermal conductivity should be amplified with a correction factor (a) that takes into account the aging of the element.

The orientation of the model building respects the orientation of the real-life building, and it is the same in each case.

The prices are only for the thermal insulation material, so the "no insulation" case does not have a price.

In figure 6 we wanted to highlight the price differences. Even if the value of the euro changes, the difference would stay the same.

The value of the heat necessary demand does not change in time because of external factors (e.g., inflation). The value would only decrease because of the aging of the material, but it would be an insignificant change.

The article contains other modifications, as we took into consideration all the suggestions received from the other reviewers. We hope you will find that the corrected version of the article addresses all suggestions.

Reviewer 2 Report

The paper compared the differences of the life cycle cost for 14 different types of thermal insultation material. The following is comments.

Line100  climate zone II for heat-100 ing and climate zone III for cooling.

please introduce what is  climate zone II  and climate zone III.

LIne 103,  Figure1 is vague and should be replaced by clear one

The most content of the paper introduce the characteristic of heat insulation material.

It is suggested to reduce discription and organize their parameters into tables.

The simulation of EnergyPlus is brief.  It should be introduced carefully.

The calculating process of life cycle cost is rough.

How to determine the different parameter of the formula (1) for different material?

Line 357-Line 382 describe the process. It is suitable for the process is introduced in the format of flow chart

Whether construction costs and fire control are considered in the life cycle cost analysis? If not, is the analysis result based on life cycle cost reasonable?

Author Response

Thank you for your time to do the review and also thank you for your recommendations as they consist in objective ideas meant to improve our work. We took into consideration your suggestions and made appropriate changes to the article as you will see in the word document as we have used Track Changes function.

Regarding the climate zones, the building is located in climate zone II for heating and climate zone III for cooling. In the Romanian normative C107/3-2005 are defined five climate zones for heating calculus and four climate zones for cooling calculus. In figure 2.3 the heat flux values were introduced along with the temperatures. The U-value measurement procedure is described in chapter 2.4.

We replaced figure 1, so it could be more comprehensible.

In chapter 3, we introduced at the end of the chapter a new table that includes the thermal insulation material’s characteristics.

We added more notes regarding the EnergyPlus simulation and, also two new figures from the modelling phase.

The calculus of the life cycle cost is described in chapter 5 and it follows formula (1). The steps used are described after the formula and we followed those precise steps using an excel file for calculation. Formula (1) is used for each material in each solution and the difference regarding the parameters for each solution are the initial cost, because of the different material price and mounting price, and the annual cost because of the different heat necessary demand. The process is introduced in a flow chart.

Construction cost is not taken into account in the life cycle cost because we have analysed an existing building. Fire control is also not considered in the life cycle cost. The only elements considered are the initial cost of the investment, the waste value of the building and the annual cost that includes: the energy cost, the maintenance cost which takes into consideration the building’s systems that maintains it in optimal parameters. Hazard factors are not taken into account, because of the very low percentage chance of happening. If it would be considered, no modernization of the buildings would be suitable.

The article contains other modifications, as we took into consideration all the suggestions received from the other reviewers. We hope you will find that the corrected version of the article addresses all suggestions.

Round 2

Reviewer 2 Report

Fig.10 is illustrated in the conclusion.

It is suggested that no figure and table in the conclusion.

Author Response

Dear Review, 

Thank you for you time reviewing our work. We have moved Figure 10 in the Discussions section in order not to appear in the Conclusions.

We look forward to your answer!

The authors
